# Factors influencing the utilisation of free-standing and alongside midwifery units in England: a qualitative research study

Denis Walsh  ,[1] Helen Spiby,[1] Christine McCourt,[2] Celia Grigg,[1] Dawn Coleby,[1] Simon Bishop,[3] Miranda Scanlon,[2] Lorraine Culley,[4] Jane Wilkinson,[5] Lynne Pacanowski,[6] Jim Thornton[7]

[1]School of Health Sciences, University of Nottingham, Nottingham, UK
[2]School of Health Sciences, City, University of London, London, UK
[3]Business School, University of Nottingham, Nottingham, UK
[4]School of Applied Social Science, De Montfort University, Leicester, UK
[5]West Cheshire CCG, Chester, UK
[6]Guys & St Thomas NHS Foundation Trust, London, UK
[7]Faculty of Medicine & Health Sciences, University of Nottingham, Nottingham, United Kingdom

**Correspondence to**
Dr Denis Walsh;
denis.walsh@ntlworld.com

## ABSTRACT

**Objective** To identify factors influencing the provision, utilisation and sustainability of midwifery units (MUs) in England.

**Design** Case studies, using individual interviews and focus groups, in six National Health Service (NHS) Trust maternity services in England.

**Setting and participants** NHS maternity services in different geographical areas of England Maternity care staff and service users from six NHS Trusts: two Trusts where more than 20% of all women gave birth in MUs, two Trusts where less than 10% of all women gave birth in MUs and two Trusts without MUs. Obstetric, midwifery and neonatal clinical leaders, managers, service user representatives and commissioners were individually interviewed (n=57). Twenty-six focus groups were undertaken with midwives (n=60) and service users (n=52).

**Main outcome measures** Factors influencing MU use.

**Findings** The study findings identify several barriers to the uptake of MUs. Within a context of a history of obstetric-led provision and lack of decision-maker awareness of the clinical and economic evidence, most Trust managers and clinicians do not regard their MU provision as being as important as their obstetric unit (OU) provision. Therefore, it does not get embedded as an equal and parallel component in the Trust's overall maternity package of care. The analysis illuminates how implementation of complex interventions in health services is influenced by a range of factors including the medicalisation of childbirth, perceived financial constraints, adequate leadership and institutional norms protecting the status quo.

**Conclusions** There are significant obstacles to MUs reaching their full potential, especially free-standing midwifery units. These include the lack of commitment by providers to embed MUs as an essential service provision alongside their OUs, an absence of leadership to drive through these changes and the capacity and willingness of providers to address women's information needs. If these remain unaddressed, childbearing women's access to MUs will continue to be restricted.

## Strengths and limitations of this study

► The richness and breadth of data captured across multiple case study sites with contrasting organisational characteristics.

► The focus groups generated discussion and insight unlikely to be obtained by individual interviews and were particularly effective in comparing service user perspectives with provider perspectives from within the same case.

► We were unable to get access to Trust documentation regarding midwifery unit policies and organisation which may have helped triangulate data from the interviews and focus groups.

► We were not able to include service users from all communities in the focus groups as we did not have translation services.

## INTRODUCTION

Since 1993, maternity care policy in England has promoted women's choice of place of birth. This became the national choice guarantee in Maternity Matters in 2007[1] which stipulated women should have three options: birth in a hospital (obstetric unit or OU), birth in two types of midwifery unit (MU)—either alongside (AMU) or free-standing (FMU)—or birth at home. MUs are birthing facilities for 'low risk' women run by midwives, though in the English context unlike other parts of the world, very few provide continuity of carer through all phases of maternity care. AMUs are attached to existing hospitals with OUs while FMUs are geographically separate. The Birthplace in England cohort study[2] reported that outcomes for low risk pregnant women were better and costs reduced if birth occurred in MUs, both AMUs and FMUs, rather than OUs. For example, having a baby in an MU reduced caesarean section rates by two-thirds, while there was no difference in

BMJ

adverse neonatal outcomes. These findings have since been supported by a systematic review of international evidence, which drew similar conclusions.[3]

The most recent National Institute for Health and Care Excellence (NICE CG190) guideline on intrapartum care therefore recommend MUs for low risk women, that is, women without significant health risk factors who are predicted to have a normal labour and birth.[4] Sandall and colleagues' research suggests this could be around 45% of all birthing women by the onset of labour.[5] Therefore, 36% of this group could be expected to give birth in MUs, allowing for a 20% intrapartum transfer rate found in the Birthplace in England study.[2] However, despite the advantages of MUs, the National Audit Office (NAO) found in 2013 that only 11% of women gave birth in those settings while the vast majority continue to give birth in OUs.[6] In addition, MUs were not equally distributed across the country.[6] A third of National Health Service (NHS) Trusts had no MUs, and those that did, were frequently underutilised with less than 10% of all births occurring in them. If 20% of births occurred in MUs, savings to the NHS maternity budget could be around £85 million/year, projecting from average cost differences.[7] This represents a 3% saving on the current annual budget of £2.6 billion for maternity care.[8]

The NICE intrapartum guidelines and maternity care policy emphasis on patient or consumer choice are in line with the direction of national policy across wider areas of healthcare. MUs could be considered an example of a complex health service 'intervention' that is a change in organisation models, based on best clinical evidence, that require a systemic, multistakeholder approach to implementation. A range of prior studies have highlighted challenges in the implementation of health policies and evidence of this nature.[9–11]

There has been no specific research investigating the reasons for the highly varied rates of MU provision across England since publication of the NAO survey. Our research was conducted to explore the reasons for these anomalies in the provision of MUs in England. The principal objectives of the study were to describe the configuration, organisation and operation of MUs in England and identify key barriers to the uptake of MU care. A three-phase mixed-methods study incorporating a Mapping Survey (phase 1), Comparative Case Studies (phase 2) and a Stakeholder Workshop (phase 3) was undertaken to answer these objectives. The national mapping of MUs and OUs nationally (including numbers and organisation) has already been reported.[12]

The most significant finding of the mapping phase (which included all 134 NHS Trusts providing maternity services) was that although the percentage of births in MUs has increased from 5% to 14% since the Birthplace in England study, that growth has occurred in AMUs.[12] This falls well short of the potential percentage of births in MUs of 36%, previously mentioned. The mapping phase also identified organisational processes within maternity services regarding MU access and utilisation. Two key

findings were, first that 97% of AMU midwives and 50% of FMU midwives were moved regularly during shifts, usually to the OU. Staff shortage or 'capacity issues' on the OU were the primary reason given for MU closures, which occurred for 28% of AMUs and 39% of FMUs. Thus, some MU midwives were providing care for low risk women in OUs, while AMUs and FMUs stood vacant. AMUs that were underutilised (ie, <10% of births) were closed three times as frequently as AMUs where>20% of women gave birth. Second, AMU admission rates were facilitated in some settings by maternity services operating an opt-out policy that is, women who met eligibility criteria were defaulted to them unless they opted otherwise, rather than a more traditional OU opt-out policy. Of the high-performing Trusts with AMUs, 73% had an opt-out policy compared with only 14% of the low-performing Trusts with AMUs.

Here we report the methods and findings for phase 2 of our overall study. The objective was to identify factors influencing the provision, utilisation and sustainability of MUs, and to understand in more depth the picture obtained in the mapping survey.[12]

## METHODS

We conducted qualitative case studies to understand and compare maternity services with different levels of progress in the implementation of MUs. Based on our mapping survey findings of 97 AMUs and 61 FMUs in England, we chose six case-study sites to study in-depth. Two were high-performing (our definition: MUs achieving 20% or more of all local facility-based births), two were low-performing (MUs achieving 10% or less of all local births) and two sites had no MUs. From 82 of the 134 Trusts meeting these eligibility criteria, in addition, we chose a mix of metropolitan and rural areas from different geographical areas with varying sizes of service and configurations. Data collection from each site involved: individual interviews with senior managerial and clinical midwives, obstetricians and neonatologists, Trust CEOs, commissioners and service user representatives in each case study site (n=57); 13 focus groups with clinical midwives (n=60); 13 focus groups with women who had recently used maternity services (n=52). Local heads of midwifery assisted the researcher in the identification of Trust clinical and managerial leadership, who were approached by the researchers directly. The midwifery leaders also facilitated the distribution of the invitation to participate in focus groups to their midwifery workforce. The service user representatives assisted researchers with identifying potential groups and venues to advertise the service user focus groups. Additionally, the research team independently approached community centres to advertise the groups. All participants provided written consent. Interviews and midwives focus groups were conducted by research staff, and service user focus groups were cofacilitated by research staff and a member of the project's service user reference group. Interview guides were

developed and piloted for all individual interviews and focus groups. Focus group size ranged from three to seven people. All focus groups and interviews were recorded and professionally transcribed.

The women's focus groups were analysed by open coding, followed by thematic distillation as outlined by Braun and Clarke.[13] All remaining focus groups and interviews were analysed with the Consolidated Framework for Implementation Research (CFIR), which provides a list of domains previously found to impact on the process of implementing evidence at an organisational level across healthcare organisations.[14] CFIR utilises five domains[15]: (1) the 'outer' wider health system (policies) and society (norms), (2) the characteristics of the individuals involved (beliefs, preferences), (3) the 'inner' context of the relevant organisations that is, NHS Trusts—their culture, networks, etc, (4) the context and nature of the 'intervention'—in this case MUs and (5) the process of change (implementing the intervention). Each of these domains has a number of constructs which findings were mapped to. Though this process is largely abductive that is, moving iteratively between inductive and deductive modes, the CFIR accommodates the creation of additional constructs inductively from the data. On completion of this, analysis proceeded by comparing and contrasting themes from the women's focus groups with the CFIR constructs 'within cases' and then on a 'cross-case' basis. Cross-case analysis was guided by the over-arching question of why some services were successful in opening, utilising and sustaining MUs and others were not.

Following analysis, we convened a meeting attended by 56 stakeholders from across England comprising provider, commissioner, education and service user constituencies, for phase 3. Findings were presented, and discussion groups identified a set of priority actions to help services to increase the provision and uptake of MUs. The detail of this phase is not reported here.

### Public and patient involvement
Public involvement was integrated into the study throughout all phases including project design, implementation, management and dissemination. One of the coinvestigators was a service user and contributed to the original idea for the research and to developing the research protocol. Four service users were recruited to a service user reference group from an established local service user maternity network. This group reviewed all aspects of the study design, including the study documents. Group members advised on approaches to achieve recruitment of women into focus groups, and cofacilitated the women's focus groups, with a member of the research staff, at the six case study sites. They also copresented the preliminary findings at the Stakeholder Workshop and cofacilitated group discussions at this event. They will also be involved in the dissemination of findings via their Facebook groups.

Additional aspects of the methods, more detail on the analytical approach across all three phases, reflections on the utility of the CFIR, sample sizes and composition will be available electronically in the Final National Institute for Health Research Report to be published on .[16]

## FINDINGS
The case study analysis distilled key themes that need addressing if English maternity services are to maximise the provision, utilisation and sustainability of MUs and therefore accrue their clinical benefits. This synthesis of the analysis will be reported under the various domains of the CFIR. Table 1 is illustrative of the process.

### Outer setting
We found strong institutional and societal pressure (risk and litigation policies, fiscal constraints) to maintain OUs as the core focus of maternity care, positioning MUs as a lesser priority and an optional extra. This involved a number of elements, including legal and governance frameworks, professional hierarchies and resource flows, which contributed to the dominance of OU care. Particularly important were perceptions of appropriate approaches to managing risk, present in the responses of representatives from all professional groups, which had not been adjusted in the light of the clinical evidence.

> There's also the potential clinical risks of people giving birth in those areas (AMUs). And we had an unfortunate death about three years ago… (Obstetrician)

> There might be a degree of fear that if people started saying that, you can go in there (to the MU), you are constantly reminded that women have to be told the risks. …because of the litigation now. (Midwife in focus group)

No professionals raised concerns about the increased risk of medical interventions associated with women giving birth in OUs.

Factors in the 'outer setting' of midwifery could be seen as contributing to a 'medical' view of childbirth that shaped perceptions of where birth should be situated. This was highlighted in women's focus groups:

> …we've been become accustomed to birth taking place in hospital (OUs) and to step outside that model you've got to face your family and peers and actually have a good reason why you want to birth outside that accepted model…hospital is perceived as safest, the 'just in case' option…

Another factor to emerge from interviews, especially from service providers, was budget constraints. Financial cutbacks within Trusts were mentioned across all sites as frustrating the development of MUs:

> I think the whole financial situation within the Trust at the moment is a driver. … Unfortunately, all our finance team will only see is the figure at the bottom of the page. …it is a sort of finance driven organisation

**Table 1** Themes mapped to CFIR domains

**Key cross-cutting themes mapped on to CFIR framework**

| CFIR domains and linked constructs | | Cross-cutting themes | | | | | |
|---|---|---|---|---|---|---|---|
| I. Intervention characteristics | | Culture and beliefs about the intervention | Resources and priorities | Organisation | Staffing | Leadership | Change |
| A | Intervention source | | | | | | |
| B | Evidence strength and quality | ■ | | | | | ■ |
| C | Relative advantage | ■ | | | | | ■ |
| D | Adaptability | | | ■ | | | |
| E | Trialability | | | | | | |
| F | Complexity | | | ■ | | | |
| G | Design quality and packaging | | | | | | |
| H | Cost | ■ | ■ | ■ | ■ | | |
| **II. Outer setting** | | | | | | | |
| A | Patient needs and resources | | ■ | | | | |
| B | Cosmopolitanism | | | | | ■ | |
| C | Peer pressure | ■ | | | | | |
| D | External policy and incentives | | ■ | ■ | | | |
| **III. Inner setting** | | | | | | | |
| A | Structural characteristics | | | ■ | ■ | ■ | |
| B | Networks and communications | ■ | | ■ | ■ | ■ | ■ |
| C | Culture | ■ | | | | | ■ |
| D | Implementation climate | ■ | | | | | |
| 1 | Tension for change | ■ | | | | | |
| 2 | Compatibility | ■ | ■ | ■ | | | |
| 3 | Relative priority | ■ | | ■ | | | |
| 4 | Organisational incentives and rewards | ■ | | | | ■ | ■ |
| 5 | Goals and feedback | ■ | | | | | |
| 6 | Learning climate | ■ | | ■ | | | |
| E | Readiness for implementation | | | ■ | ■ | ■ | |
| 1 | Leadership engagement | | | ■ | ■ | ■ | |
| 2 | Available resources | | ■ | ■ | ■ | | |
| 3 | Access to knowledge and information | | | | | | |
| **IV. Characteristics of individuals** | | | | | | | |
| A | Knowledge and beliefs about the intervention | ■ | ■ | | | | |
| B | Self-efficacy | | | | | | |
| C | Individual stage of change | ■ | | | ■ | | |
| D | Individual identification with organisation | | | | ■ | | |
| E | Other personal attributes | | | | | | |
| **V. Process** | | | | | | | |
| A | Planning | | ■ | ■ | | | ■ |
| B | Engaging | | ■ | ■ | | | ■ |
| 1 | Opinion leaders | | | | | | ■ |
| 3 | Champions | | | | | | ■ |
| 4 | External change agents | | | | | | ■ |
| C | Executing | | ■ | ■ | | | ■ |
| D | Reflecting and evaluating | | | | | | ■ |

CFIR, Consolidated Framework for Implementation Research.

and you're forever trying to find ways of saving money, cutting costs, etc (Midwifery Manager)

All respondents appeared to accept the need for Trusts to save money as a 'fait accompli' and the unaffordability of MUs as a 'fact' as typified by the phrase '*we're in a period of austerity now*' and positioned maternity as competing and losing out to other services. The findings indicated little awareness of the evidence on cost-effectiveness of MU facilities.

## Characteristics of individuals

Closely related to a medicalised view of childbirth, we found mixed beliefs among individuals about the efficacy of MUs, with pockets of strong scepticism across high and low uptake sites. In many instances, these attitudes took precedence over opposing views emanating from the clinical evidence. Antipathy towards MUs was particularly strong in the case of FMUs, in relation to which several common assumptions were noted. These included the perceived superior safety of the medical model, that FMUs and AMUs offer essentially an identical service and that FMUs are not popular with women:

> I think majority of women and all my friends will opt for an alongside MU, because most women do want the option of midwifery led but if anything goes wrong they just want to go down that corridor, through that door. (Midwifery Manager)

Many midwives, especially in sites with no MUs, were reported as actively resisting the development of an FMU:

> …they (the midwives) were completely horrified at the idea of having a standalone midwifery-led unit. (Midwifery Manager)

While variations of this attitude could be found across all sites, within high-performing sites we did find existing AMU and FMU 'champions' who saw themselves as contributing to the 'mission' or 'vision' of midwifery led birth:

> The vision is to up the numbers, so we have the quality boards, and we are aiming to increase the deliveries in the midwifery led unit, and home deliveries. … we are continuously striving to increase it. (Midwife)

## Inner setting

We found that collaboration between MUs and OUs was important for the successful embedding of the MU model, and pockets of excellent collaborative relations were reported within high-performing sites. More commonly, this did not occur, creating an 'us and them' atmosphere as illustrated by this segment of a focus group transcript between an FMU midwife and the facilitator:

> Int: I went on a transfer, and the reception I got was non-existent.
>
> Fac: What do you mean?

> Int: There was nobody waiting…there wasn't a cot in the room, no midwife came, I had to find somebody.
>
> Fac: But they're always told ahead that you're coming?
>
> Int: Oh yeah, they know you're coming. I've been greeted with 'oh, here comes another failure from FMU'.

We also found evidence in some Trusts of a culture of marginalising and undervaluing of FMUs. As a result, several FMUs were under threat of closure, even in high-performing sites. The two dominant rationales for closure were that they are underused and too expensive as illustrated by these quotes:

> Well it (FMU) is small and we do have to understand how viable it (FMU) is because you can't spread yourself so thin. So it (FMU) is difficult to manage because we're covering so many other areas, and the birth rates numbers are very low. (Manager)

> If you spoke to any of the consultants I am sure they would say it (FMU) should be closed because it's a waste of money. And it's an unfair allocation of resources, in a relatively resource poor environment. (Manager)

In addition, we found evidence that FMUs can be subjected to a mixture of managerial neglect and authoritarian control from their host Trusts. An FMU manager said:

> They (Trust management) always pay us lip service… don't promote us'…we've been fighting for a year to get a video on the Trust website, of a tour of our birth centre…. You do feel like the poor relation. (Focus Group Midwife)

This manifested in several contradictory messages coming from some Trusts. We found examples of all of the following: using FMUs to solve capacity crises at times of peak activity while threatening them with closure at other times; restricting opportunities for FMU staff to promote their services as illustrated in the quote above; FMU staff not being consulted on strategic changes that impacted on them as this excerpt from an individual interview revealed:

> To hear the news about the closure (of the FMU) on the TV rather than from the organisation was terrible, so it makes them, you know, lose confidence. (FMU Manager).

## Intervention characteristics

Embedding MU provision was perceived as presenting a number of challenges. MUs are intended to provide care to low risk women, where midwives can practice the skills of normal midwifery. However, a number of midwife respondents felt that practicing within them required different skills and a level of confidence, which they were not well prepared for.

Because everyone has worked in such a high-risk environment, you become deskilled to an extent, and feel a bit apprehensive about normal birth… you know, trusting that women can have babies low risk. (Focus Group Midwife)

Midwifery managers and midwives in our study recommended mandatory training in normal birth skills to address this concern. Linked to a perceived deficit in skills and arguably more influential was a lack of confidence among midwives to make decisions in MU settings where midwives are more autonomous, as illustrated from this midwife focus group:

One of the effects that that has had, is that it has decreased a lot of the midwives' confidence in this unit, of providing low risk care, because they don't have the environment, a consistent one, in which to do it properly…. when you're on labour ward you become over reliant on the doctors.

### Process

There were considerable differences across sites in the process of implementing and developing MUs. Leadership emerged as key to successful implementation.

it's crucial to have an inspirational leader. If you don't have somebody at the very top who is passionate about it (MUs) happening, it won't happen. And they must cascade, get everybody onboard. (Midwives Focus Group)

a charismatic leader to kind of bring it together… unless you've got that then I think it's quite hard to bring it to fruition. (Manager)

Continuity of leadership contributing to organisational memory was also stressed:

I think the birth centres are being used less at the moment, and that does seem to coincide with a change of leadership. (Midwives Focus Group)

Only a few sites had an active policy of the ongoing promotion of MUs to their local women to increase and sustain their utilisation:

So you have to do a lot of positive promotion, you have to get out there. And you're almost selling a product. And that's how we saw it. So we did lots of promotional events, and got lots in the press, about the opening of the FMU. (Manager)

Successful implementation was also dependent on a clear clinical pathway from the beginning of pregnancy until the onset of labour. For example, there was a wide variation in the information women had, or were given, about MUs—within and between Trusts. The majority of women in the focus groups reported not receiving information. Participants from five of the six case study Trusts mentioned not being given information about the local MUs (including the two which have more than 20% of women giving birth in a MU).

Well it's just that nobody gave us the information about it (MUs). That's the main thing. I didn't know nothing about it. I didn't even know it even existed. (Women's Focus Group)

Women expressed concerns about the place of birth booking process, such as whether it was necessary to decide at the beginning of pregnancy, how it was done, and if it was possible to change your mind.

I wasn't aware that you had to decide before you went in for your booking appointment, so I was asked on the spot and I didn't know. But the midwife said that you have to choose now because they have to book the hospital in advance. (Women's Focus Group)

### DISCUSSION

This research has illuminated why MUs are underused in England and still not available in many NHS Trusts. The central challenge in all case study sites was introducing and sustaining what was still perceived as an alternative configuration (MUs) into an existing and embedded mainstream, 'taken for granted' model (OUs) which has been in place for the past 5 decades. OUs are the default place of birth for the vast majority of women in England, regardless of women's risk profiles. Utilising the domains of the CFIR, our findings show how several multiple external (outer context), and internal (inner context) factors, alongside personal beliefs of key players, intrinsic features of MU services and the process of change itself combine to reinforce the status quo and slow the growth of MUs.

Coxon *et al*[17 18] and Scamell[19] argue that the construction of birth as risky in policy initiatives and by service providers over recent decades has shaped women's preference for birth in OUs. Birth as a risky endeavour is a by-product of the medicalisation of childbirth over a similar period that has seen caesarean section rates rise exponentially.[20 21] As Coxon demonstrated, if women's first experience of birth is in a hospital labour ward, they are likely to choose the same for subsequent births.[17] What this study has illustrated is that professional perceptions of what is risky and how risk should be managed can be out of step with evidence—in this case, the evidence on the safety of different birth settings.[2 3]

Despite national guidelines based on extensive evidence recommending MUs for women at low obstetric risk, we found that managers, midwives and clinicians in provider settings harboured considerable ambivalence about the safety of MUs. Research has shown that personal belief can moderate evidence[22] and is a key variable to address in systematic reviews of what facilitates the translation of evidence into practice.[23 24] FMUs were especially vulnerable to negative beliefs about their efficacy even though they pre-date AMUs by decades, although under the title

of maternity homes or general practitioner units. Though AMUs are a relatively new phenomenon, there has been an exponential increase in their use over the past 6 years, even if still at a low overall level, which could reflect the broader bias favouring the embedded system of OUs as AMUs are colocated with them.

Financial constraints within Trusts were often seen as limiting the development of MUs. While economic evaluations suggest the overall economic outcomes of increasing births in MUs is positive,[25] the start-up costs were seen as a barrier, and the longer term savings from lower morbidity in the target population that accrue across the health system were not recognised. In a climate of scarcity, new ways of structuring care must demonstrably save money, or at least, be perceived to, in the short term.

A defining characteristic of MUs as an intervention is that their functioning is entirely dependent on midwives because they are midwife-led and managed. Skills in managing normal labour and birth and decision-making autonomy are prerequisites for practise in this setting. Our findings highlighted a lack of midwifery confidence and skill that can be traced back to the training and practice of UK midwives within predominantly obstetric-led services. Numerous surveys and papers have demonstrated this over the last 30 years since Robinson's pioneering research on the loss of traditional midwifery skills in the 1980s[26–29].

Our findings pinpoint the importance of leadership to the process of managing organisational change of this magnitude. Best *et al*'s realist review of large system transformation of health services[30] found that blending designated leadership (someone in charge of the programme) with distributed leadership (professionals/partner organisations sharing responsibility for delivering it) was the most successful at embedding and sustaining change. For the successful development and operationalisation of MUs, leadership needs to be exercised vertically via the layers of organisational hierarchy and horizontally across different professional groups; and at each of these levels, designated leadership and distributive leadership should be combined. An important component of leadership was the identification and subsequent impact of having 'champions'. Champions of MUs were either clinicians, managers or service users who were highly influential in promoting the service and recruiting support for it. Designated leaders working with champions were better at establishing clinical pathways for women to optimise access and utilisation of MUs. This included user-friendly information to promote the choice of MUs, adopting an opt-out mechanism for AMUs and employing a consultant midwife to oversee and develop MUs.

A final issue illuminated by this study was the finding that despite arguments posited by service managers in relation to lack of demand, the majority of women in our focus groups reported lack of awareness of these services and lack of information provision about their options. This echoes the findings of Rayment *et al* in relation to women's access to MUs in England[31] and Henshall *et al*'s

systematic review, which highlighted professionals lack of skills and confidence in providing information, in a context where such services continue to be viewed as alternative.[32]

Our findings help explain the difficulty moving away from the existing status quo. Under each of the domains of the CFIR, the study identified issues that would appear to slow the growth of MUs. The current constitution of healthcare organisations, the policy environment, aspects of training, as well as complexities in the nature and process of change together work to maintain the dominance of OUs for birth. The study findings address the specific challenges for maternity care but also illuminate wider issues relevant to implementation science in health.

The strength of this comparative case study methodology is the richness and breadth of data captured across multiple sites with differing organisational characteristics. In addition, focus groups generated discussion and insight unlikely to be obtained by individual interviews. They were particularly effective in comparing service user perspectives with provider perspectives from within the same case. Inevitably, we were not able to include a full range of service users in the focus groups as we did not have translation services. Despite this, we did have Black, Asian and Minority Ethic representation in some of the focus groups.

## IMPLICATIONS

The key implication of this research is that, in many areas of England, women at low risk of complications do not have access to the maternity care that evidence shows is most suitable for them, because of the factors highlighted in this paper.

The importance of leadership was a principal finding from our case studies as it is a critical factor in the normalisation of an intervention to the point where it is no longer appraised as marginal but becomes incorporated and understood as a core part of the service.[33]

It was clear from our study that inequality of access to information is primarily a matter of women not being given information about the option of MU care. Having an opt-out policy for FMUs should also be explored. FMUs have the additional advantage of being a more local provision for some women and therefore meeting the wider health service principle of moving care closer to home.[34] In addition women may benefit from a higher quality of information about place of birth options and evidence, provided at different stages of the pregnancy.[35]

The increase in the overall number of MUs since 2010 is due to the opening of AMUs. Trusts also need to value their FMU(s) as central to the broader maternity service provision and an important choice for low risk women. In particular, the common perception that FMUs are a financial burden unless operating at maximum capacity needs to be challenged as the available evidence suggests that they are cheaper than supporting the same women to birth in an OU, even when the MU is operating at

around 30% capacity. This is because health economists factored in the savings they generate in reduced intervention and maternal morbidity.[7 25] FMU facilities could also be used more extensively for other outpatient services and could arguably operate as part of a community hub as envisioned by the Implementing Better Births policy document.[36 37]

## CONCLUSIONS

NICE Intrapartum Care Guidance (CG190) first recommended birth in MUs for low risk women in 2014, but their potential for women across England is not being realised. This is because of the challenge of embedding them into the existing hospital-based OU model that has been in place for several decades. Changing the status quo requires leadership from both commissioners and providers and a clearly articulated belief in the value of MUs, exercised through committing resources, streamlining care pathways and ongoing promotion to service users.

**Acknowledgements** We are extremely grateful to all who contributed to the study, including Heads of Midwifery for their support to the stage 1 mapping phase; all who contributed during through interviews or focus groups to the stage 2 case study phase and participants at the stage 3 stakeholder event. We would also like to thank the members of the Service User Reference Panel for their invaluable contributions: Melissa Thomas, Leanne Stamp, Samantha Foulke and Joanne Whyler. We would like to thank Dr Juliet Rayment for her contribution to the media analysis of free-standing midwifery units that had closed and 'Which? Birth Choice' for allowing us access to their data. We would also like to acknowledge and thank Sheila Adamson at the University of Nottingham for all her work as the Research Secretary over 24 months.

**Contributors** DW: chief investigator, associate professor in Midwifery, Denis. walsh@ntlworld.com, principal author; HS: coinvestigator, professor in Midwifery, Helen.spiby@nottingham.ac.uk; CMC: coinvestigator, professor in Maternal Health, Christine.mccourt.1@city.ac.uk; CG: research fellow, celia.grigg@xtra.co.nz; DC: early career academic fellow dawn.coleby@dmu.ac.uk; SB: coinvestigator, senior lecturer in Business School, Simon.Bishop@nottingham.ac.uk; MS: coinvestigator, Service User, Miranda.Scanlon@city.ac.uk; LC: coinvestigator, Emeritus Professor of Social Science & Health, lac@dmu.ac.uk; JW: GP Commissioner, jane.wilkinson14@nhs.net; LP: head of Midwifery, homeofthepacs@gmail.com; JT: professor of Obstetrics, Jim.Thornton@nottingham.ac.uk. All authors performed drafting and revising content critically for important intellectual content, substantial contribution to the interpretation of data, final approval of the version to be published, agreement to be accountable for all aspects of the work in ensuring that questions related to the accuracy or integrity of any part of the work are appropriately investigated and resolved.

**Funding** This work was supported by the National Institute for Health Research (NIHR) Health Services and Delivery Research programme (Ref: 14/04/28).

**Competing interests** Professor Thornton reports being a member of the HTA and EME Boards. Dr Scanlon reports grants from NIHR, during the conduct of the study; personal fees from 'WHICH?', grants from NIHR, personal fees from National Perinatal Epidemiology Unit, personal fees from Rod Gibson Associates, personal fees from Midwifery Unit Network, outside the submitted work.

**Patient consent for publication** Not required.

**Ethics approval** Ethical approval was granted for phase 2 of the study the West Midlands—Solihull Research Ethics Committee (IRAS ID 200356) as phases 1 and 3 were deemed service development.

**Provenance and peer review** Not commissioned; externally peer reviewed.

**Data availability statement** Data are available in a public, open access repository. Data Sharing Statement: Data are available in a public, open access repository: https://www.journalslibrary.nihr.ac.uk/programmes/hsdr/140428/#/

**ORCID iD**
Denis Walsh http://orcid.org/0000-0002-5435-6403

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
