## [Reviewer comments · BMJ Open]

ARTICLE DETAILS

TITLE (PROVISIONAL)	Factors influencing the utilisation of free-standing and alongside midwifery units in England: A Qualitative Research Study
AUTHORS	Walsh, Denis; Spiby, Helen; McCourt, Christine; Grigg, Celia; Coleby, Dawn; Bishop, Simon; Scanlon, Miranda; Culley, Lorraine; Wilkinson, Jane; Pacanowski, Lynne; Thornton, Jim

VERSION 1 – REVIEW

REVIEWER	Prof dr JJHM Erwich Head of Obstetrics Chair of the Hospital Adverse Incidents Committee UMCG (CCP) Department of Obstetrics and Gynecology University Medical Center Groningen University of Groningen Homepostcode CB20 PO Box 30 001 9700 RB Groningen The Netherlands Phone +31 50 3613008 tracer 43016 E-mail: j.j.h.m.erwich@umcg.nl
REVIEW RETURNED	15-Oct-2019

GENERAL COMMENTS	I have to congratulate the authors on this work and have no suggestions for improvement. It shows again that implementation of changes in health care is difficult. The identification of several barriers hopefully will waken the field to get on going! I suggest this to be in the regular BMJ with an editorial supporting the findings. However, between the lines the clear opinion of the authors is noticable, and together with the firm conclusions, people might feel attacked (although rightly so) and see these conclusions as a crusade, in the end perhaps doing more harm than good. This should be weighed by the authors, and perhaps a few lines added to say that no one will frustrate these developments deliberately (prevent loss of face). but knowing now more details, specific barriers needs to be addressed, Who is going to check on any progress?
--

REVIEWER	Caroline Homer Burnet Institute, Australia
REVIEW RETURNED	15-Nov-2019

GENERAL COMMENTS	Thank you for the opportunity to review this paper which details the findings from a qualitative component of a wider study about midwife
---

	units in England. The study includes six case sites and collected data through interviews and focus groups. The abstract needs clarification. How many people were involved in the study? The section under Participants that explains the site selection should be in the setting I would think. I am confused about the sub-heading – Intervention: Establishing MUs. My understanding is that these MUs were already established, just not being used as well as they might. I am not convinced that this is an intervention study per se. The Introduction was not very clear in places. To a general reader, what a AMU and a FMU needs to be explained. Para 3 starts with “these changes in guidelines” but it is not clear which guidelines or policies are being referred to. What is NIHR for a non UK reader? Is it necessary to name the funder in the Introduction? Para 4 mentions that highly varied rates of MU provision but this is not explained. What was the variation? How highly varied was it? What is the model of midwifery care in the MUs. Is there midwifery continuity of care provided? The mapping phase findings at the end of page 6 onto page 7 needs a citation. Where are these results found? The end of the Introduction there needs to be a clear aim of this part of the project – the aim of this paper is to At the moment is not clear whether the objective is for this paper or phase 2 or perhaps they are the same. The objective in the Abstract is not really the same as what is reported at the end of the Introduction. A better lining up of the aim is needed. The methods need more detail. How were the sites selected – other than being high or low performing? What was the available number of MUs from which the 6 were selected? I was surprised that 20% was selected as being high performing. This seems rather low given the recommendation they cite from the Sandall study is >40%? How were participants recruited to the study? Who did the interviews and focus groups? The statement that the full methods will be available in the future is not helpful and cannot be included. If the authors want to refer to this, it must be available now – not in the future. In the presentation of the results using the framework, it might be useful to present it as a diagram. This might make the findings a bit more dynamic. The opening statement of the Results section states that these are the key themes that need addressing if maternity services are to maximise the clinical, psychological, workforce and economic benefits of MUs. This is quite different to the objective of the study in the Abstract and in the Introduction. Better alignment is needed. I agree that the data collected would be rich it is just that did not come across in the findings. A large amount of data were collected – 57 interviews, 13 focus groups with midwives and 13 focus groups with women and finally a workshop with 56 stakeholders. It feels as if only a tiny amount of data were presented and at not a very deep level or well synthesised. The analysis feels limited and things are left hanging, for example, cost, fear, access etc. Much of the findings
--	---

	just feel like a series of quotes without a lot of synthesis or abstraction. That may be as the CFIR framework was used rather than an inductive analytical approach but I felt reading the paper that much was lost by doing this. The discussion is interesting but it is hard to see clear links with what is presented in much of the Discussion with what was actually presented as findings. I am not convinced that the study contributes to understanding use of the CFIR framework – it is a way of presenting the information but I am not sure that it really helped identify the organisational changes issues in depth. The paper needs to reflect upon, and include a section on the limitations of the study and the approach. In summary, the study is really important and essential in the UK and other countries. A deeper analysis needs to be presented and a clearer link between the findings and the discussion needs to be made.
--	---

VERSION 1 – AUTHOR RESPONSE

1. SRQR checklist included with page/line numbers from manuscript
2. Interventions section removed from Abstract
3. Participant numbers now in Abstract
4. Now called 'Qualitative Research Study' in title and throughout manuscript
5. Sentence added regarding the development of interview guides and we have now included more description of the focus groups
6. We have addressed the points from the 2nd reviewer regarding improving the Abstract
7. In regard to feedback from Reviewer 1, we have softened the language in various sections, including the abstract so that our opinion should appear less polemical and arising directly from the data. We have not added any comment 'to say that no one will frustrate these developments deliberately' as we feel this makes an assumption that this has already happened and our data does not support this. But we do believe we have made an unequivocal challenge to providers to address the under-utilization of MUs, FMUs in particular.
8. Regarding the 2nd reviewer, as stated, we have addressed the issues with the Abstract
9. We have addressed all the issues in the Introduction e.g. definition of MUs, acronyms, funder name removed, a fuller explanation of the optimum usage of MUs. Paragraph 2 of the Introduction gives the results of the National Audit Office Report which surveyed MUs, showing that they were not widely available throughout England, and, where they did exist, their utilisation was well short of their potential capacity. A sentence has been added regarding the paucity of continuity models associated with MUs. Citation for the mapping paper was included (last sentence of 4th paragraph)
10. The Objective in Abstract has now been aligned to the objective stated at the end of the introduction
11. We have added text regarding Methods, answering queries made by the reviewer included more on how sites were selected, the pool from which they were selected, participants were recruited and who did the interviews. Our definition of high-performing was based on births occurring in MUs which, extrapolating from Sandall's figures, could be maximally expected to be 36% (see middle of 2nd

paragraph of Introduction). We have now stated that full details of methods will be publicly available from 31st of January, 2020 with a reference when the full report will be published.

12. We have now included a Table to show how women focus group themes were mapped to CFIR domains and their constituent constructs.

13. The opening sentence in the Results section has been changed to reflect the research objective.

14. Thank you for your feedback regarding the analysis. Because of words limits, we chose to not explicate in detail the analytical process and had to be highly selective about results we reported on. However, after reflecting on your feedback, we have elaborated on the analysis to help bridge the gap between data, interpretation and synthesis. We have also moved some sections of the results that are more resonant with each other and moved some discussion sections to reflect this sequential change. We hope this adds to coherence and flow. We have rewritten parts of the discussion so that it links better with the Findings.

15. We have removed the reflection on the utility of CFIR.

16. A strengths and limitations section follow the Abstract and there is also a paragraph about this immediately before the Implications Heading.

VERSION 2 – REVIEW

REVIEWER	Caroline Homer Burnet Institute, Australia
REVIEW RETURNED	19-Dec-2019
GENERAL COMMENTS	The authors have responded to my comments. It would have been more helpful to the reviewer if the Response to Reviewers document was presented in a way that addressed each of the reviewer concerns in turn rather than a more global approach as was presented. Nonetheless, I am happy to recommend that the paper is accepted.